# Poultry farmer response to disease outbreaks in smallholder farming systems in southern Vietnam

Alexis Delabouglise[1,2]*, Nguyen Thi Le Thanh[3], Huynh Thi Ai Xuyen[4], Benjamin Nguyen-Van-Yen[3,5], Phung Ngoc Tuyet[4], Ha Minh Lam[3,6], Maciej F Boni[1,3,6]

[1]Center for Infectious Diseases Dynamics, The Pennsylvania State University, University Park, United States; [2]UMR ASTRE, CIRAD, INRAE, Université de Montpellier, Montpellier, France; [3]Wellcome Trust Major Overseas Programme, Oxford University Clinical Research Unit, Ho Chi Minh City, Viet Nam; [4]Ca Mau sub-Department of Livestock Production and Animal Health, Ca Mau, Viet Nam; [5]École Normale Supérieure, CNRS UMR 8197, Paris, France; [6]Centre for Tropical Medicine and Global Health, Nuffield Department of Medicine, University of Oxford, Oxford, United Kingdom

**Abstract** Avian influenza outbreaks have been occurring on smallholder poultry farms in Asia for two decades. Farmer responses to these outbreaks can slow down or accelerate virus transmission. We used a longitudinal survey of 53 small-scale chicken farms in southern Vietnam to investigate the impact of outbreaks with disease-induced mortality on harvest rate, vaccination, and disinfection behaviors. We found that in small broiler flocks (≤16 birds/flock) the estimated probability of harvest was 56% higher when an outbreak occurred, and 214% higher if an outbreak with sudden deaths occurred in the same month. Vaccination and disinfection were strongly and positively correlated with the number of birds. Small-scale farmers – the overwhelming majority of poultry producers in low-income countries – tend to rely on rapid sale of birds to mitigate losses from diseases. As depopulated birds are sent to markets or trading networks, this reactive behavior has the potential to enhance onward transmission.

*For correspondence:
alexis.delabouglise@gmail.com

Competing interests: The authors declare that no competing interests exist.

## Introduction

Livestock production systems have been a major driver of novel pathogen emergence events over the past two decades (*Gao et al., 2013*; *Guan et al., 2002*; *Rohr et al., 2019*). The conditions enabling the emergence and spread of a new disease in the human population partly depend on human behavioral changes, like hygiene improvements or social distancing, in the face of epidemiological risks (*Funk et al., 2010*). The same observation applies to disease emergence and spread in livestock populations as farmers adapt their farm management to maximize animal production and welfare while limiting cost in a constantly changing ecological and economic environment (*Chilonda and Van Huylenbroeck, 2001*).

Poultry farming generates substantial risk for emergence of novel infectious diseases. It is now the most important source of animal protein for the human population and the industry is changing rapidly (*FAOSTAT, 2019*). The link between poultry sector expansion and pathogen emergence is exemplified by the worldwide spread of the highly pathogenic form of avian influenza (AI) due to the H5N1 subtype of influenza A, after its initial emergence in China in 1996 (*Guan et al., 2002*; *Guan and Smith, 2013*). Highly Pathogenic Avian Influenza (HPAI) causes severe symptoms in the most vulnerable bird species (including chicken, turkey, and quail), with mortality rates as high as

**eLife digest** The past few decades have seen the circulation of avian influenza viruses increase in domesticated poultry, regularly creating outbreaks associated with heavy economic loss. In addition, these viruses can sometimes 'jump' into humans, potentially allowing new diseases – including pandemics – to emerge.

The Mekong river delta, in southern Vietnam, is one of the regions with the highest circulation of avian influenza. There, a large number of farmers practice poultry farming on a small scale, with limited investments in disease prevention such as vaccination or disinfection. Yet, it was unclear how the emergence of an outbreak could change the behavior of farmers.

To learn more, Delabouglise et al. monitored 53 poultry farms, with fewer than 1000 chickens per farm, monthly for over a year and a half. In particular, they tracked when outbreaks occurred on each farm, and how farmers reacted. Overall, poultry farms with more than 17 chickens were more likely to vaccinate their animals and use disinfection practices than smaller farms. However, disease outbreaks did not affect vaccination or disinfection practices.

When an outbreak occurred, farmers with fewer than 17 chickens tended to sell their animals earlier. For instance, they were 214% more likely to send their animals to market if an outbreak with sudden deaths occurred that month. Even if they do not make as much money selling immature individuals, this strategy may allow them to mitigate economical loss: they can sell animals that may die soon, saving on feeding costs and potentially avoiding further contamination. However, as animals were often sold alive in markets or to itinerant sellers, this practice increases the risk of spreading diseases further along the trade circuits.

These data could be most useful to regional animal health authorities, which have detailed knowledge of local farming systems and personal connections in the communities where they work. This can allow them to effect change. They could work with small poultry farmers to encourage them to adopt efficient disease management strategies. Ultimately, this could help control the spread of avian influenza viruses, and potentially help to avoid future pandemics.

100% reported in broiler flocks (*OIE, 2018*). Some subtypes of AI viruses have caused infection in humans, including H5N1, H5N6, H7N9 and H9N2, with potentially severe illness and, in the cases of H7N9 and H5N1, a high case-fatality rate (*Chen et al., 2013*; *Claas et al., 1998*; *Peiris et al., 1999*; *Yang et al., 2015*). So far, reports of human-to-human transmission of these subtypes of influenza have been either absent or anecdotal, but the risk that they make the leap to a human pandemic is a persistent if unquantifiable threat to public health (*Imai et al., 2012*). While HPAI does not persist in poultry populations in most affected countries, it has become endemic in parts of Asia and Africa and is periodically re-introduced into other areas like Europe and North America (*Lai et al., 2016*; *Li et al., 2014*). In affected countries, major factors influencing HPAI epidemiology appear to be farm disinfection, poultry vaccination, and marketing of potentially infected birds through trade networks, all of which depend on farmers' management decisions (*Biswas et al., 2009*; *Desvaux et al., 2011*; *Fasina et al., 2011*; *Henning et al., 2009*; *Kung et al., 2007*).

It is still unclear how and to what extent changes in outbreak risk or mortality risk affect the behavior of poultry farmers. An anthropological study in Cambodia showed that high levels of farmer risk awareness associated with HPAI did not translate into major changes in their farming practices (*Hickler, 2007*). Qualitative investigations conducted in Vietnam, Bangladesh, China, and Indonesia reported that farmers sometimes urgently sell or cull diseased poultry flocks as a way to mitigate economic losses, but evidence of this behavior's onward epidemiological impact was not available (*Biswas et al., 2009*; *Delabouglise et al., 2016*; *Padmawati and Nichter, 2008*; *Sultana et al., 2012*; *Zhang and Pan, 2008*). Additionally, it is unknown whether poultry farmers increase application of disinfection practices or vaccination rates against avian influenza in response to disease outbreaks occurring in their flocks. Changes in farm management caused by variations in epidemiological risk have not been quantified for any livestock system that we are aware of, primarily because of the lack of combined epidemiological and behavioural data in longitudinal studies of livestock disease (*Hidano et al., 2018*). *Ifft et al., 2011* compared the evolution of chicken farm sizes and disease prevention in administrative areas with different levels of HPAI prevalence in

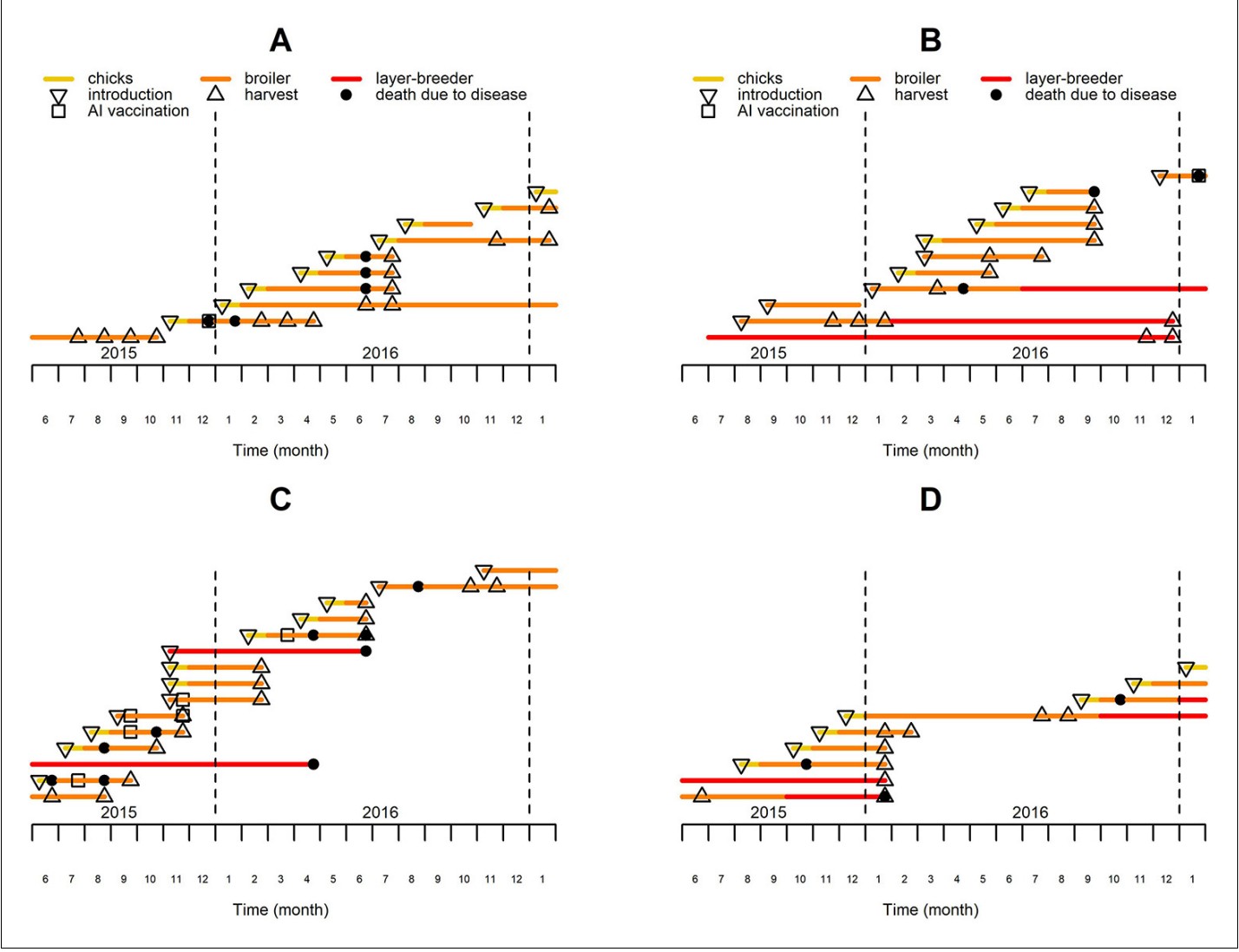

**Figure 1.** History of chicken flocks present in four of the observed farms over the study period. Each colored line represents the period over which a single chicken flock was present on the farm, with the color code indicating the production type, which may vary during the course of the flock production period. The major events affecting the flocks are located with specific symbols on the corresponding lines and months.

Vietnam, and *Hidano and Gates, 2019* modelled the effect of cattle mortality and production performance on the frequency of sales and culling in New Zealand dairy farms. One limitation of these two studies is that the dynamics were observed over year-long time steps, which does not allow for a precise estimation of the timing of farmer response after the occurrence of disease outbreaks and the potential feedback effect of this response onto the resulting outbreaks or epidemics.

Vietnam has suffered human mortality and economic losses due to HPAI. The disease has been endemic in the country since its initial emergence in 2003–2004 (*Delabouglise et al., 2017*). Small-scale poultry farming is practiced by more than seven million Vietnamese households, mostly on a scale of fewer than 100 birds per farm (*General Statistics Office of Vietnam, 2016*). In addition to HPAI, other infectious diseases severely affect this economic sector, including Newcastle disease, fowl cholera, and Gumboro, which are all endemic despite the availability of vaccines for their control (*OIE, 2019*).

We present a longitudinal study of small-scale poultry farms where we aimed to characterize the effect of disease outbreaks on livestock harvest rate (i.e. rate of removal by sale or slaughter) and on two prevention practices, vaccination and farm disinfection. This longitudinal farm survey was

conducted on small-scale poultry farms in the Mekong river delta region of southern Vietnam (*Delabouglise et al., 2019*).

## Results

Fifty three farms were monitored from June 2015 to January 2017. Monthly questionnaires were used to collect farm-level information on poultry demographics (number, introduction, death and departure of birds), mortality (cause of death, observed clinical symptoms) and management by farmers. The main poultry species kept on these farms was chicken, with ducks and Muscovy ducks as the other two primary relevant species held. Farmers kept an average number of 79 chickens, 53 ducks and 7 Muscovy ducks per farm over the 20 month study period. Each farm's poultry were classified into 'flocks', defined as groups of birds of the same age, species, and production type. *Figure 1* illustrates the farms' structure and dynamics. Broiler chicken flocks were kept for 15.5 weeks on average after which most chickens were harvested and a minority was consumed or kept on the farm for breeding and egg production (*Delabouglise et al., 2019*).

We fit mixed-effects general additive models (MGAM) with three different dependent variables: a 'harvest model' of the probability of harvesting (i.e. selling or slaughtering) chicken broiler flocks at a particular production stage (data points are flock-months), an 'AI vaccination model' of the probability of performing AI vaccination on chicken broiler flocks which had never received AI vaccination (data points are flock-months), and a 'disinfection model' of the probability of disinfecting farm facilities (data points are farm-months). Disease outbreaks were included in each model as independent categorical variables. Disease outbreaks refer to the occurrence of poultry mortality attributable to an infectious disease in the corresponding farm at different time intervals before the corresponding month. Specifically, outbreaks were defined by the death of at least two birds of the same species with similar clinical symptoms in the corresponding farm in the same month, one month prior, and two months prior. For the harvest model, only outbreaks in chickens were considered. For the AI vaccination model, outbreaks in chickens and outbreaks in any other species were included as two separate covariates. For the disinfection model, outbreaks in any of the species present in the farm were considered. In chickens, outbreaks with 'sudden deaths' (i.e. the death of chickens less than one day after the onset of clinical symptoms) are considered as being indicative of HPAI infection (*Mariner et al., 2014*). Therefore, we created two sub-categorical variables for outbreaks in chickens, with sudden deaths (OS, 'outbreaks sudden') and with no sudden deaths (ONS, 'outbreaks not sudden'). The three dependent variables are likely influenced by several other farm-, flock-, and time-related factors, justifying the inclusion of control covariates which are reported in *Table 1* and described in detail in the 'Materials and methods'.

A total of 1656 broiler chicken flock-months were available for analysis. They belonged to 391 chicken flocks present on 48 farms. In 18.8% of flock-months non-sudden outbreaks (ONS) were observed in chickens on the same farm, 1.6% of flock-months saw sudden outbreaks (OS) in chickens on the same farm, and 7.2% of flock-months saw disease outbreaks in poultry of other species on the same farm (*Table 1*). The percentages are very similar for outbreaks occurring one month prior and two months prior since they are averaged over similar sets of months, with differences mostly related to outbreak frequency in the two first months and two last months of the study period. Additional descriptive statistics on control covariates are described in *Table 1*. Out of 1656 broiler chicken flock months, 1503 flock-months were selected for the harvest analysis after excluding data points with new-born chicks and flock-months in which all the chickens had died (see Materials and methods). No harvest occurred in 995 flock-months, complete harvest occurred in 258 flock-months, and partial harvest occurred in 250 flock-months. The probability of harvest during a month, with partial harvests weighted appropriately, was 23.9%. Excluding flock-months of already vaccinated chickens (and some with missing data), 1318 flock-months were selected for the AI vaccination analysis (see Materials and methods). AI vaccination was performed in 7.5% (99/1318) of flock-months. The 99 vaccinated flocks were from 29 different farms (out of 48 farms keeping broiler chickens). For the disinfection model, 858 farm-months belonging to 52 farms were included (see Materials and methods). During 552 farm-months the farm was fully disinfected, during 259 farm-months the farm was not disinfected at all, and during 47 farm-months disinfection was performed for some (but not all) of the flocks present in the farm. The probability of disinfection during a month, with partial disinfections weighted appropriately, was 67.4%. The best fit statistical models and their parameter

**Table 1.** Summary statistics of variables.

| Continuous variable | Min | 1st quartile | Median | 3rd quartile | Max |
|---|---|---|---|---|---|
| **Broiler chicken flocks (n = 391)** | | | | | |
| Number of flocks of broiler chickens per farm | 2 | 22 | 36 | 44 | 75 |
| Number of observation months per broiler flock | 1 | 3 | 4 | 5 | 12 |
| **Broiler chicken flock-months (n = 1656)** | | | | | |
| Flock size (*n*) (number of birds) | 2 | 10 | 16 | 35 | 580 |
| Anticipated age at maturity (*t\**) (weeks) | 9.5 | 13.1 | 17.4 | 19.6 | 43.6 |
| Age at the time of observation (*t*) (weeks) | 1.6 | 6.3 | 12.3 | 19 | 53.6 |
| Difference *t*- *t\** (*δt*) (week) | −37.2 | −11.1 | −5.2 | 1 | 36.1 |
| Calendar time (*T*) | 3 | 7 | 11 | 16 | 20 |
| Proportion harvested (%) | 0 | 0 | 0 | 33.3 | 100 |
| Number of chicken flocks introduced in the same month onto the same farm | 0 | 0 | 0 | 1 | 4 |
| Number of chicken flocks introduced in the month prior onto the same farm | 0 | 0 | 0 | 1 | 2 |
| Number of broiler chickens present on the same farm in other flocks (bird) | 0 | 10 | 25 | 61 | 900 |
| Number of broiler ducks present on the same farm (bird) | 0 | 0 | 0 | 25 | 3630 |
| Number of broiler Muscovy ducks present on the same farm (bird) | 0 | 0 | 0 | 6 | 80 |
| Number of layer chickens present on the same farm (bird) | 0 | 2 | 6 | 13 | 350 |
| Number of layer ducks present on the same farm (bird) | 0 | 0 | 0 | 0 | 11 |
| Number of layer Muscovy ducks present on the same farm (bird) | 0 | 0 | 0 | 2 | 30 |
| **Farm-months (n = 876)** | | | | | |
| Number of broiler chickens (bird) | 0 | 8 | 28 | 64 | 912 |
| Number of broiler ducks (bird) | 0 | 0 | 4 | 31 | 3630 |
| Number of broiler Muscovy ducks (bird) | 0 | 0 | 0 | 6 | 80 |
| Number of layer chickens farm (bird) | 0 | 0 | 4 | 10 | 358 |
| Number of layer ducks (bird) | 0 | 0 | 0 | 0 | 500 |
| Number of layer Muscovy ducks (bird) | 0 | 0 | 0 | 2 | 30 |
| Proportion flocks farmed with disinfection (%) | 0 | 0 | 100 | 100 | 100 |

| Qualitative variable | Proportion of observations |
|---|---|
| **Broiler chicken flock-months (n = 1656)** | |
| Occurrence of outbreak with no sudden death in chickens on the same farm in the current month | 18.8% |
| Occurrence of outbreak with sudden death in chickens on the same farm in the current month | 1.6% |
| Occurrence of outbreak in other species on the same farm in the current month | 7.2% |
| Confinement indoors or in enclosure | 32.8% |
| Previously vaccinated for AI | 20.2% |
| Previously vaccinated for Newcastle Disease | 7.1% |
| **Farm-months (n = 876)** | |
| Occurrence of outbreak in any species | 23.4% |

values are summarized in *Table 2*. Fitted spline functions cannot be elegantly summarized by their coefficients and are displayed graphically in *Figures 2* and *3*.

The harvest model showed support for associations between flock- and farm-level covariates, particularly the difference between flock age and age at maturity and the probability of harvesting broiler chickens. The model explained 34.2% of the observed deviance. There was no statistical support for a temporal auto-correlation of the probability of harvest of broiler chicken flocks on a given farm (*Table 2*). As the interaction term between flock size (*n*) and outbreak occurrence was significant (p<0.01) but difficult to interpret (displayed in *Supplementary file 1*), we separated the flocks

**Table 2.** Fitted parameters of the broiler chicken flock harvest and AI vaccination and farm disinfection models.

| Model | Variable | | | Odds-ratio (with 95% CI) | p-value |
|---|---|---|---|---|---|
| Harvest | Flock size $\leq$ 16 chickens | ONS chickens* | Same month | 2.06 (1.23; 3.45) | $<10^{-2}$ |
| | | | −1 month | 2.06 (1.17; 3.62) | 0.02 |
| | | | −2 months | 0.41 (0.19; 0.92) | 0.03 |
| | | OS chickens** | Same month | 9.34 (2.13; 40.94) | $<10^{-2}$ |
| | | | −1 month | 0.18 (0.01; 4.95) | 0.32 |
| | | | −2 months | 0.88 (0.15; 5.04) | 0.89 |
| | | Number of broiler chickens in the farm (square root) | | 1.05 (1; 1.11) | 0.06 |
| | | combined effect of the difference between current age and age at maturity ($\delta t$) and the age at maturity ($t^*$) (spline transformation) | | *Figure 2* | $<10^{-3}$ |
| | Flock size > 16 chickens | OS chickens** | Same month | 1.02 (0.23; 4.46) | 0.98 |
| | | | −1 month | 3.89 (0.82; 18.46) | 0.09 |
| | | | −2 months | 3.1 (0.51; 18.77) | 0.22 |
| | | Number of broiler chickens in the farm (square root) | | 1.05 (1; 1.11) | 0.05 |
| | | combined effect of the difference between current age and age at maturity ($\delta t$) and the age at maturity ($t^*$) (spline transformation) | | *Figure 2* | $<10^{-3}$ |
| AI vaccination | | Outbreak chickens | Same month | 0.75 (0.29–1.92) | 0.55 |
| | | | −1 month | 0.78 (0.29–2.11) | 0.63 |
| | | | −2 months | 0.27 (0.08–0.89) | 0.04 |
| | | Outbreak others | Same month | 4.62 (1.08–19.72) | 0.04 |
| | | | −1 month | 0.51 (0.09–2.89) | 0.45 |
| | | | −2 months | 0.42 (0.06–2.91) | 0.39 |
| | | Number of broiler chickens in the farm (square root) | | 0.92 (0.82–1.03) | 0.2 |
| | | Number of broiler Muscovy ducks in the farm (square root) | | 0.74 (0.57–0.96) | 0.03 |
| | | Number of layer ducks in the farm (square root) | | 2.95 (1.15–7.57) | 0.03 |
| | | Number of layer Muscovy ducks in the farm (square root) | | 1.9 (1.07–3.36) | 0.03 |
| | | Confinement | | 24.6 (6.32–95.6) | $<10^{-3}$ |
| | | Proportion harvested | | 0.01 (0–0.37) | 0.02 |
| | | Spline transform of the logarithm of the flock size ($n$) | | *Figure 3A* | $<10^{-3}$ |
| | | Spline transform of the logarithm of the flock age ($t$) | | *Figure 3B* | $<10^{-3}$ |
| | | Spline transform of the calendar time ($T$) | | *Figure 3C* | $<10^{-3}$ |
| Disinfection | | Number of broiler Muscovy ducks in the farm (square root) | | 1.07 (1.01–1.13) | 0.02 |
| | | Number of layer ducks in the farm (square root) | | 1.25 (1.02–1.53) | 0.04 |
| | | Number of layer chickens in the farm (square root) | | 1.3 (1.12–1.51) | $<10^{-3}$ |
| | | Spline transform of the calendar time ($T$) | | *Figure 3D* | $<10^{-3}$ |

\* ONS: Outbreak with no sudden deaths.

\*\* OS: Outbreak with sudden deaths.

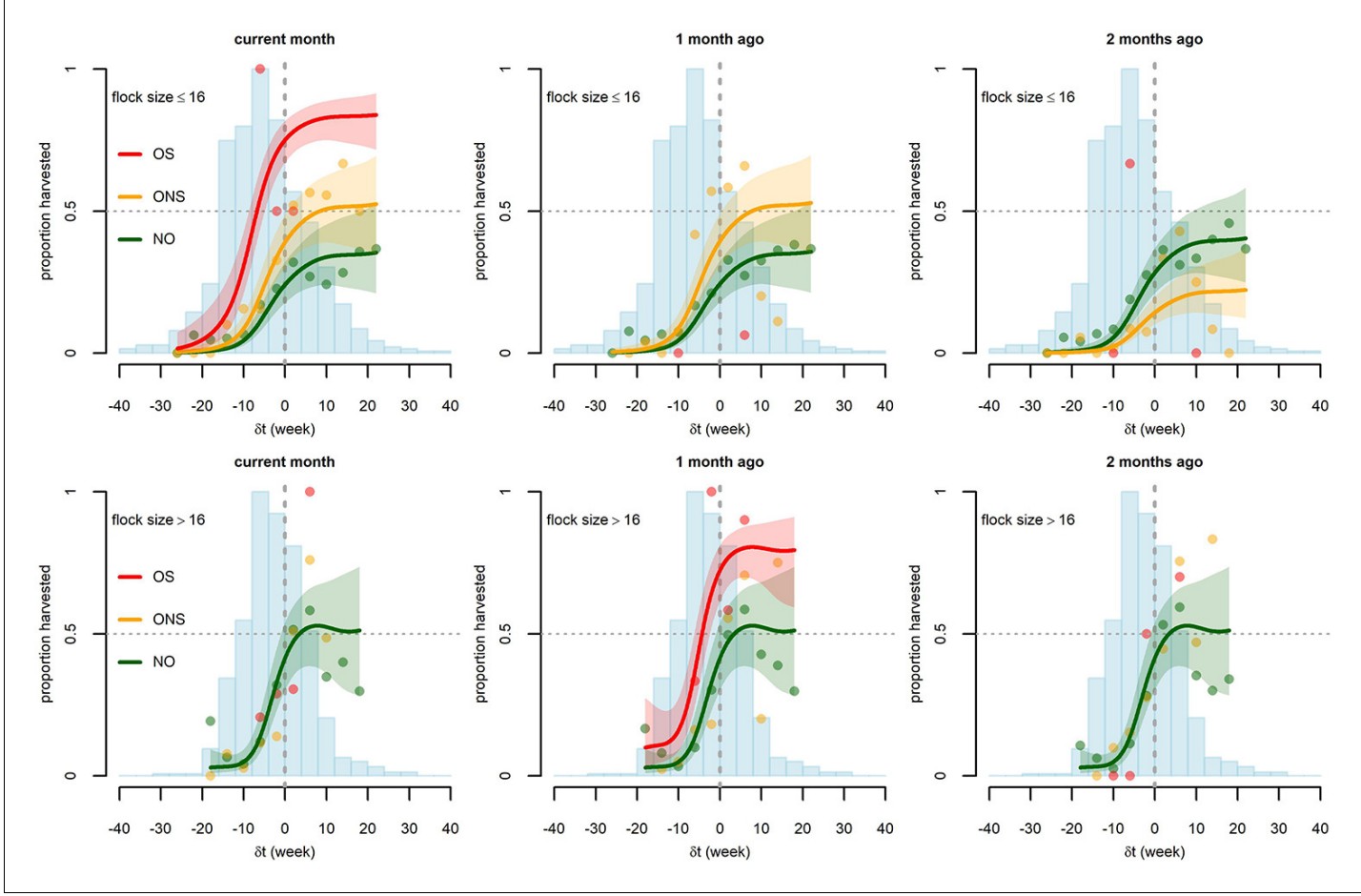

**Figure 2.** Graphical representation of the relationship between the difference $\delta t$ (current flock age - flock age at maturity) and the proportion of broiler flocks harvested in the absence (NO, green) or presence of outbreaks with disease-induced mortality, either with sudden deaths (OS, red) or with no sudden deaths (ONS, orange). Three different outbreak timings are considered: same month (left), one month prior (middle), and two months prior (right). Two different classes of flock size are considered: small,<17 chickens (top) and large,≥17 chickens (bottom). Points are the observed proportions (estimated from at least two flock-months) and lines are the predictions of the fitted Harvest model, along with 90% confidence bands. Model predictions with outbreaks are only displayed when fitted outbreak effects have some statistical significance (p<*0.10*) (see *Table 2*). Blue histograms correspond to the number of observed flock-months in the different classes of $\delta t$ (scaled to their maximum, 139 in the top graphs and 157 in the bottom graphs).

The online version of this article includes the following figure supplement(s) for figure 2:

**Figure supplement 1.** Graphical representation of the relationship between the difference $\delta t$ (current flock age - flock age at maturity) and the proportion of broiler flocks harvested in the absence (green color - NO) or presence of outbreaks with disease-induced mortality (dark orange color).

into large and small. A threshold value of 16 birds per flock gave the lowest Akaike Information Criterion (AIC) (when using a categorical variable indicating small flock or large flock), and flocks of 16 birds or fewer (52% of all flocks) were designated as small while flocks of 17 or more (48% of all flocks) were designated as large. As expected, the probability of harvest was found to be strongly dependent on the difference ($\delta t$) between the flock age and the anticipated age at maturity, with older flocks being more likely to be sold. The probability of harvest was close to zero when $\delta t < -15$ weeks, i.e. flocks that are more than 15 weeks away from maturity. The probability of harvest increased steeply from $\delta t = -10$ to $\delta t = 0$. For $\delta t > 0$ (flocks past their age at maturity), the probability of harvest was consistently high but lower than 100% and did not depend on age. Larger flocks had a steeper increase in harvest probability as a function of $\delta t$; once past the age at maturity ($\delta t > 0$), the estimated probability of harvest for large flocks was higher (interquartile range: 41–61%) than for small flocks (interquartile range: 30–41%) (*Figure 2*).

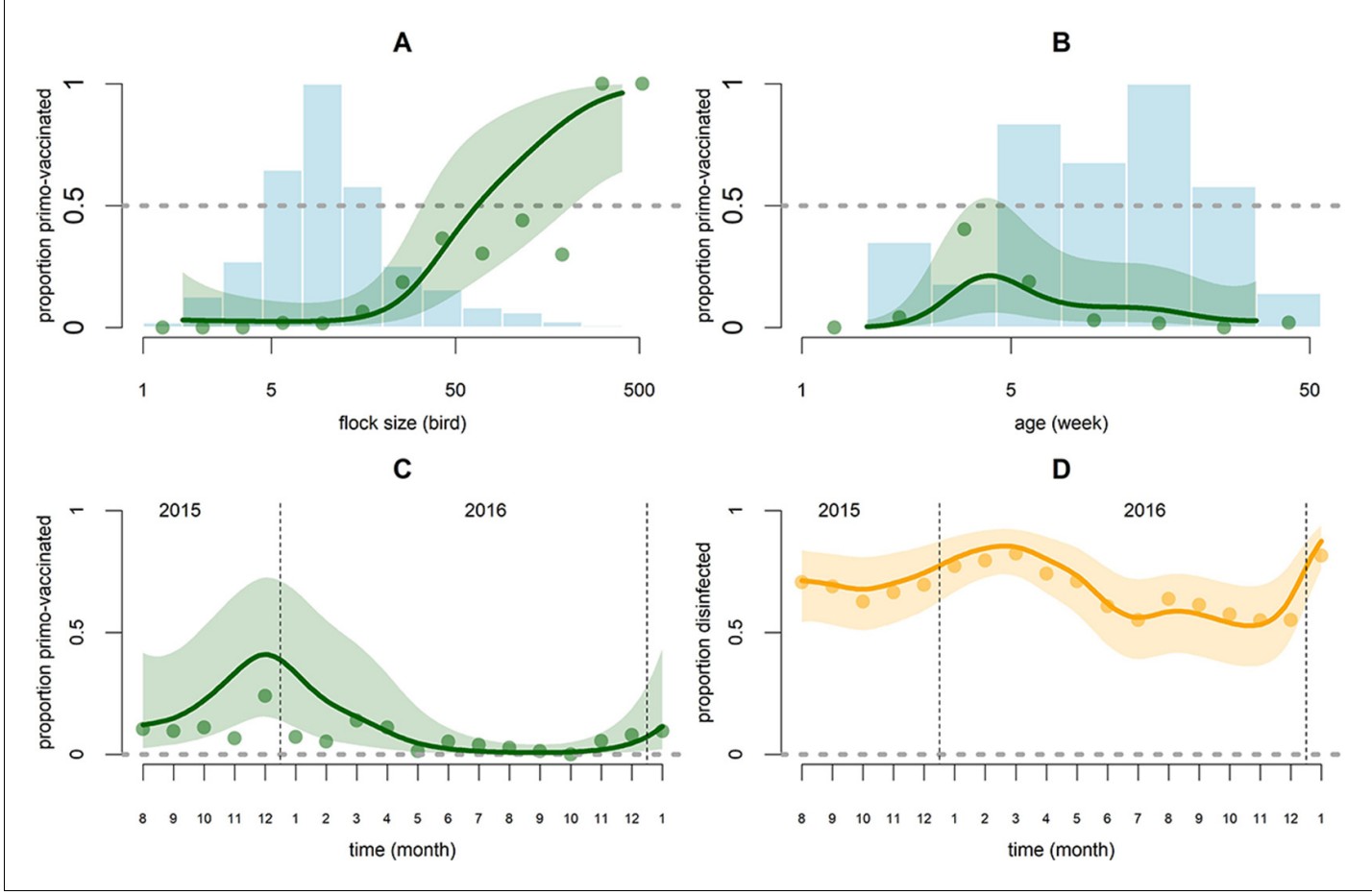

**Figure 3.** Graphical representation of predictions of the AI vaccination and disinfection models as functions of covariates whose effect is modeled with thin plate smooth splines. For the AI vaccination model (green) these covariates are flock size (n) (A), age (t) (B) and calendar time (T) (C). For the disinfection model (orange), the covariate is calendar time (T) (D). Points are the observed proportions and lines are the predictions along with the 90% confidence band. In graphs C and D the proportions are displayed on the logit scale. Blue histograms correspond to the number of observed flock-months in the different classes of $log(n)$ (A) and t (B) (scaled to their maximum, 402 in A and 345 in B).

Disease outbreaks substantially affected the likelihood of harvest of broiler chickens. The probability of harvest of small flocks was significantly higher on farms that had experienced a non-sudden outbreak (ONS) in chickens in the same month (odds ratio (OR) = 2.06; 95% confidence interval (CI): 1.23–3.45) or the previous month (OR = 2.06; 95% CI: 1.17–3.62) and was lower on farms that had experienced an ONS in chickens two months prior (OR = 0.41; 95% CI: 0.19–0.92). The probability of harvest of small flocks was much higher on farms that had experienced a sudden outbreak (OS) in the same month (OR = 9.34: 95% CI: 2.13–40.94). We used the fitted model to predict the mean harvest proportion in the study population with and without outbreak. Estimated mean harvest proportions of small flocks were 17% (no outbreak), 28% (ONS), and 56% (OS) when considering outbreaks occurring in the same month; this corresponded to harvest increases of 56% and 214% for ONS and OS outbreaks, respectively. Estimated mean harvest proportion was 18% (no outbreak) and 28% (ONS) when considering outbreaks one month prior; this corresponded to a 56% increase in harvest in case of ONS one month prior. Mean harvest proportions were 20% (no outbreak) and 11% (ONS) when considering outbreaks two months prior, indicating a 47% decrease in harvest in case of ONS two months prior. For large flocks, ONS in chickens (in any month current or previous) did not have any effect on the harvest of broiler chickens (the removal of ONS variables decreased the model AIC). The occurrence of OS in chickens one month prior may be positively associated with early harvest with an estimated 76% increase in harvest proportion (OR = 3.89; 95% CI: 0.82–18.46; p=0.09). However, we do not have sufficient statistical power to support this association. In the last six

months of data collection, farmers were asked to indicate the destination of harvested birds. Based on these partial observations, flocks harvested during or one month after outbreaks in chickens (OS or ONS) were more likely to be sold to traders and less likely to be slaughtered at home (*Table 3*). The likelihood of harvest was also positively correlated with the number of other broiler chickens present on the farm (*Supplementary file 1*, p < 0.01). It was not found to be affected by the concomitant introduction of other flocks, vaccination status, or calendar time (*T*). The farm random effect was significant for large flocks ($\sigma$ = 0.74; 95% CI: 0.47–1.17) and not significant for small flocks.

The number of outbreaks with sudden deaths is relatively small (11 small flock-months and 14 large flock-months occurred on farms experiencing an OS in the same month) and OS are potentially subject to misclassification, depending on how regularly farmers check on their chickens. Therefore, in order to ensure the robustness of our result, we conducted a separate analysis with merged OS and ONS categories. The results are displayed in *Supplementary file 2* and *Figure 2—figure supplement 1*. The probability of harvest of small flocks was significantly higher on farms that had experienced an outbreak in chickens in the same month (Odds ratio (OR) = 2.34; 95% CI: 1.43–3.81) or the previous month (OR = 1.96; 95% CI: 1.14–3.37) and was lower in farms that had experienced an outbreak in chickens two months prior (OR = 0.45; 95% CI: 0.22–0.92). For large flocks, there was no statistical support for outbreaks in chickens having an effect on the harvest of broiler chickens.

The AI vaccination model showed support for an effect of flock size on vaccination, while explaining 71.9% of the observations' deviance. The likelihood of broiler chicken vaccination against AI strongly increased with flock size; probability of vaccination was almost zero for flocks of 16 birds or fewer and nearly 100% for flocks of more than 200 birds (*Figure 3A*). Vaccination was preferentially performed at 4.3 weeks of age (*Figure 3B*). Flocks kept indoors or in enclosures had a substantially higher chance of being vaccinated than flocks scavenging outdoors (OR = 24.6; CI: 6.32–95.6). Harvested flocks were less likely to receive an AI vaccination (OR = 0.01; CI: 0–0.37). The likelihood of AI vaccination was dependent on calendar time: it increased over the September-January period and decreased during the rest of the year (*Figure 3C*). There was no statistical support for a temporal auto-correlation of the probability of vaccination of broiler chicken flocks against AI on a given farm. The farm random effect was significant ($\sigma$ = 2.86; CI: 1.88–4.35). We failed to obtain convergence when fitting the specific effects of OS and ONS in chickens, so we used an aggregate variable 'outbreak in chickens' instead (*Table 2*). Broiler chicken flocks were more likely to be vaccinated if an outbreak had occurred in the same month in other species (OR = 4.62; CI: 1.08–19.72; p=0.04) and less likely to be vaccinated if an outbreak had occurred two months prior in chickens (OR = 0.27; CI: 0.08–0.89; p=0.03). These two effects were weakly significant and should be interpreted with caution (*Table 2*). The coefficients for interaction terms between outbreak occurrence and flock size were not significantly different from zero. The number of broiler Muscovy ducks present in the farm had a negative effect (p=0.03) and the number of layer ducks and layer Muscovy ducks had a positive effect (both p=0.03) on the probability of AI vaccination (*Table 2*).

The disinfection model showed evidence that larger farms were more likely to report routine disinfection of their premises; the model explained 61.9% of the observations' deviance. Probability of disinfection on farms was auto-correlated in time (likelihood ratio test for 1 month AR-model on

**Table 3.** The destination of harvested broiler chicken flocks with or without occurrence of outbreaks of disease-induced mortality in chickens of the same farm in the same month or one month prior (%).

| Destination | No outbreak | Outbreak with no sudden death (ONS) | Outbreak with sudden death (OS) |
|---|---|---|---|
| Sale to traders | 28% | 45% | 45% |
| Sale at market | 5% | 16% | 0% |
| Sale to other farmers | 2% | 3% | 0% |
| Sale unspecified | 12% | 4% | 11% |
| Slaughter at home | 36% | 20% | 11% |
| Gift | 5% | 8% | 11% |
| Feed farmed pythons | 5% | 1% | 22% |
| Other | 7% | 3% | 0% |

residuals; p<0.0001); this was not observed for the harvest or vaccination models (both p>0.3). Consequently, the disinfection model was improved by fitting an AR-1 autoregressive model using the 'gamm' routine of the 'mgcv' R package. The estimated AR-1 autoregressive coefficient was high ($\rho$ = 0.71). The likelihood of disinfection of farm facilities increased with the number of layer-breeder hens (OR = 1.3; CI: 1.12–1.51; p=0.001), layer-breeder ducks (OR = 1.25; CI: 1.02–1.53; p=0.03), and to a lesser extent broiler chickens (OR = 1.07; CI: 1.01–1.13; p=0.02) present on the farm (*Table 2*). Farm disinfection appeared to have a seasonal component. It was least likely in October-November and most likely in the January-April period (*Figure 3D*). It was not found to be affected by the occurrence of outbreaks (no decrease in AIC when including outbreak occurrence).

## Discussion

Regions like the Mekong river delta combine high human population density, wildlife biodiversity, and agricultural development. As such, they are considered hotspots for the emergence and spread of novel pathogens (*Allen et al., 2017*). The high density of livestock farmed in semi-commercial operations with limited disease prevention practices further increases the risk of spread of emerging pathogens in livestock and their transmission to humans (*Henning et al., 2009*). In-depth studies of poultry farmers' behavioral responses to disease occurrence in animals are needed to understand how emerging pathogens – especially avian influenza viruses – may spread and establish in livestock populations and how optimal management policies should be designed. To the best of our knowledge, this study is the first to provide a detailed and quantified account of the dynamics of livestock management in small-scale farms and its evolution in response to changing epidemiological risks shortly after disease outbreaks occur. While our analysis was performed on a geographically restricted area, the decision-making context of the studied sample of farmers is likely to be applicable to a wide range of poultry producers in low- and middle-income countries. Small-scale poultry farming, combining low investments in infrastructure, no vertical integration, and subject to limited state control on poultry production and trade, is common in most regions affected by avian influenza, in Southeast Asia, Egypt, and West Africa (*Burgos et al., 2008a*; *Hosny, 2006*; *Obi et al., 2008*; *Sudarman et al., 2010*). Additional longitudinal surveys using a similar design should be carried out in other countries and contexts to assess the presence or absence of the behavioral dynamics observed here.

In our longitudinal study, owners of small chicken broiler flocks resorted to early harvesting of poultry, also referred to as depopulation, as a way to mitigate losses from infectious disease outbreaks. The revenue earned from the depopulation of flocks might be low, either because birds are still immature or because traders use disease symptoms as an argument to decrease the sale price. Nevertheless, depopulation allows the farmer to avoid a large revenue loss resulting from disease-induced mortality or the costs of management of sick or dead birds. More importantly, farmers avoid the cost of feeding chickens at high risk of dying and prevent the potential infection of subsequently introduced birds. Our results also suggest that the depopulation period, which lasts approximately two months, is followed by a 'repopulation' period during which farmers lower their harvest rate, possibly to increase their pool of breeding animals in order to repopulate their farm.

The epidemiological effect of chicken depopulation is likely twofold: on the one hand it may slow the transmission of the disease on the farm, since the number of susceptible and infected animals is temporarily decreased (*Boni et al., 2013*); on the other hand, since most poultry harvested during or just after outbreaks were sold to itinerant traders or in markets, depopulation increases the risk of dissemination of the pathogens through trade circuits (*Delabouglise and Boni, 2020*). There is epidemiological evidence that poultry farms can be contaminated with HPAI through contact with traders who purchase infectious birds and that infectious birds can contaminate other birds at traders' storage places and in live bird markets (*Biswas et al., 2009*; *Fournié et al., 2016*; *Kung et al., 2007*). Overall, chicken depopulation may reduce local transmission at the expense of long-distance dissemination of the pathogen. The rapid sale of sick birds also exposes consumers and actors of the transformation and distribution chain (traders, slaughterers, retailers) to an increased risk of infection with zoonotic diseases transmitted by poultry, like avian influenza (*Fournié et al., 2017*). Large flocks appear to be less readily harvested upon observation of disease mortality. Farmers may depopulate large flocks only upon observation of sudden deaths, but the number of observations in our study is too small to demonstrate statistical significance of this effect. The likely reason for this

difference is that the sale and replacement of larger flocks incurs a higher transaction cost. While small flocks are easily collected and replaced by traders and chick suppliers in regular contact with farmers, the rapid sale of larger flocks probably requires the intervention of large-scale traders or several small-scale traders with whom farmers have no direct connection, and who may offer a lower price per bird. When farm production increases, farmers tend to rely on pre-established agreements with traders, middlemen, or hatcheries on the sale dates in order to reduce these transaction costs, giving them little possibility to harvest birds at an earlier time (*Catelo and Costales, 2008*).

The timing of harvest of broiler chickens is also affected by farm-related factors, as shown by the significance of the farm random effect in large flocks. Indeed, farmers have different economic strategies, some aiming at optimizing farm productivity and harvesting broilers as soon as they reach maturity, and others using their poultry flocks as a form of savings and selling their poultry whenever they need income or when prices are high (*ACI, 2006*). For the latter category, the sale of chickens presumably depends on variables which were not captured in this study, like changes in market prices, economic shocks affecting the household, a human disease affecting a member of the household, or celebrations. Those variables should be captured in future surveys in order to improve the predictive power of harvest models. Another limit of the model is the use of a proxy of the chicken weight combining age, age at maturity, and flock size, rather than the actual weight, which is difficult to monitor in a longitudinal study of this size.

While government-supported vaccination programs have been proposed as a suitable tool to control AI in small scale farms with little infrastructure (*FAO, 2011*), in this survey AI vaccination was almost exclusively performed in large flocks kept indoors or in an enclosure. Vaccination against AI is believed to be inexpensive for farmers as vaccines are supplied for free by the sub-department of animal health of Ca Mau province and performed by local animal health workers. However, vaccination may still involve some fixed transaction cost as farmers have to declare their flocks to the governmental veterinary services beforehand. Also it is possible that small flocks, being less likely to be sold to distant larger cities (*Tung and Costales, 2007*), are less likely to have their vaccination status controlled, making their vaccination less worthwhile from the farmers' perspective. Crucially, it is these smaller flocks that are more likely to be sold into trading network during outbreaks. Finally, farmers' willingness to expand their production, invest in farm infrastructure, and implement AI prevention are likely correlated. Farms with a large breeding-laying activity tend to invest more in preventive actions (disinfection and vaccination) compared to farms specialized in broiler production. This may reflect a higher individual market value of layer-breeder hens compared to broiler chicks, making their protection more worthwhile.

While vaccination against AI and disinfection appear to depend on individual farmer attitude, as shown by the significance of the farm random effects, they still vary over time when viewed across all farms (*Figure 1*). Contrary to harvesting behavior, these preventive actions have a seasonal component (*Figure 3C and D*) indicating a willingness to maximize the number of vaccinated broiler chickens and the protection against other diseases during the January-March period. The January-March period is the period of lunar new year celebrations in Viet Nam, commonly associated with higher poultry market prices and an increased risk of disease transmission, as has been observed for avian influenza (*Delabouglise et al., 2017*; *Durand et al., 2015*). In response, farmers tend to invest more in disease prevention practices at this time and veterinary services provide more vaccines and disinfectant for free. Farm disinfection has a significant temporal autocorrelation component and is unaffected by disease outbreaks, indicating that farmers are slower at adapting this practice to changing conditions. Some events may affect the frequency of vaccination and disinfection on a long time frame. For example, the peak in AI vaccination observed at the end of 2015 can be interpreted as a part of a long-term response to the high HPAI incidence reported in early 2014 (*Delabouglise et al., 2017*). The time period of the present study is too short to provide a statistical support for these long term dynamics.

The data from this study were recorded at farm level on monthly basis, which limits the risk of recall bias. It was an easy task for farmers participating in the survey to report the number of deaths and associated clinical symptoms. We cannot, however, totally exclude the risk of misclassification of disease outbreaks, especially the misclassification of outbreaks in chickens as 'sudden', as it is influenced by the frequency of inspection of chickens flocks by farmers and other members of the households.

The main result of the study is that, as poultry flock size decreases, farmers increasingly rely on depopulation rather than preventive strategies to limit economic losses due to infectious diseases. In the current context, depopulation mainly results in the rapid transfer of potentially infected chickens to trade systems, increasing the risk of pathogen dissemination. In response, governments may use awareness campaigns directed at actors of poultry production systems to communicate information on the public health risks associated with the trade of infected birds. However, if the economic incentives for depopulating are high enough, communication campaigns may fail to produce noticeable results. Small-scale farmers could play an active role in the control of emerging infectious diseases if they were given the opportunity to depopulate their farm upon disease detection without disseminating pathogens in trade circuits, as theoretical models predict that depopulation can maintain a disease-free status in farming areas (*Delabouglise and Boni, 2020*). Policymakers may be able to encourage the establishment of formal trade agreements enabling and encouraging 'virtuous' management of disease outbreaks in poultry. For example, in some areas of Vietnam, poultry originating from farms experiencing disease outbreaks are partly used as feed for domestic reptiles (farmed pythons and crocodiles) or destroyed with the support of larger farms (*Delabouglise et al., 2016*).

The last 23 years of emerging pathogen outbreaks and zoonotic transmissions failed to prepare us for the epidemiological catastrophe that we are witnessing in 2020. Multiple subtypes of avian influenza viruses have crossed over into human populations since 1997 (*Gao et al., 2013*; *Lai et al., 2016*), all resulting from poultry farming activities. Small-scale poultry farming is likely to be maintained in low- and middle-income countries as it provides low-cost protein, supplemental income to rural households, and is supported by consumer preference of local indigenous breeds of poultry (*Burgos et al., 2008a*; *Epprecht, 2005*; *Sudarman et al., 2010*). If we ignore the active role that poultry farmers play in the control and dissemination of avian influenza, we may miss another opportunity to curtail an emerging disease outbreak at a stage when it is still controllable.

## Materials and methods

### Data collection

An observational longitudinal study was conducted in Ca Mau province in southern Vietnam (*Delabouglise et al., 2019*; *Thanh et al., 2017*) with the collaboration of the Ca Mau sub-Department of Livestock Production and Animal Health (CM-LPAH). Fifty poultry farms from two rural communes were initially enrolled and three additional farms were subsequently added to the sample in order to replace three farmers who stopped their poultry farming activity. The two communes were chosen by CM-LPAH based on (1) their high levels of poultry ownership, (2) their history of HPAI outbreaks, and (3) likelihood of participation in the study (*Thanh et al., 2017*). Study duration was 20 months, from June 2015 to January 2017. Monthly Vietnamese-language questionnaires were used to collect information on (1) number of birds of each species and production type, (2) expected age of removal from the farm, (3) number of birds introduced, removed, and deceased in the last month, (4) clinical symptoms associated with death, (5) vaccines administered, (6) type of poultry housing used, and (7) disinfection activity. Each farm's poultry were classified into 'flocks', defined as groups of birds of the same age, species, and production type (*Delabouglise et al., 2019*). Because individual poultry cannot be given participant ID numbers in a long-term follow-up study like this, a custom python script was developed to transform cross-sectional monthly data into a longitudinal data set on poultry flocks (*Nguyen-Van-Yen, 2017*).

Recruitment was designed to have a mix of small (20–100 birds) and large (>100 birds) farms and a mix of farms that were 'primarily chicken' and 'primarily duck'. As multiple poultry species were present on most farms, the chicken and duck farm descriptors were interpreted subjectively. The enrollment aim was to include 80% small farms among chicken farms and 50% small farms among ducks farms; there was approximately equal representation of chicken and ducks farms, but many could have been appropriately classified as having both chickens and ducks. As the residents in the two communes were already familiar with CM-LPAH through routine outreach and inspections, all invitees agreed to study participation. The farm sizes and poultry compositions were representative of small-scale poultry ownership in the Mekong delta regions, but other potential selection biases in the recruitment process could not be ascertained. No sample size calculation was performed for the

behavioral analysis presented here, as we had no baseline estimates of sale patterns or disease prevention activities. The duration and size of the study was planned to be able to observe about 1000 poultry flocks (all species and production types included).

## Selection of observations

For the 'harvest model' and 'AI vaccination model', we focused our analysis on broiler chicken flocks, since chicken was the predominant species in the study population, the overwhelming majority of chicken flocks were broilers, and their age-specific harvest was easier to predict than the harvest of layer-breeder hens. Additionally, only six layer-breeder chicken flocks were vaccinated against AI during the study period. Observations made in the two first months of the study were discarded since, during these two months, it was unknown whether farms had previously experienced outbreaks.

In the 'disinfection' model, observations were farm-months. A total of 876 farm-months were available for inclusion in the model. We removed farm-month with missing data on disinfection performed by farmers (18 farm-months) so 858 farm-months were used to fit the disinfection model. In the 'harvest' and 'AI vaccination' models, observations were chicken broiler flock-months. We selected all chicken flock-months more than 10 days old at the time of data collection and classified by farmers as 'broilers'. A total of 1656 flock-months were available for inclusion in the model. In the "harvest model we removed flock-months which were less than 20 days old at the time of data collection. This 20 day threshold was chosen because some newborn flocks below this age were partly sold, not for meat consumption but for management on other farms. Also, we removed flock-months where no chickens were available for harvest because they had all died in the course of the month (25 flock-months). In total, 153 flock-months were removed and 1503 flock-months were used to fit the harvest model. In the 'AI vaccination' model, we removed flock-months of flocks which had already been vaccinated against avian influenza in a previous month, since vaccination is usually performed only once (among the 338 vaccinated flocks, only eight were vaccinated a second time). We also removed flock-months whose housing conditions were not reported (four flock-months). In total, 338 flock-months were removed and 1318 flock-months were used to fit the AI vaccination model.

## Selection of covariates

A disease outbreak was defined as the death of at least two birds of the same species – on the same farm, in the same month, with similar clinical symptoms – as this may indicate the presence of an infectious pathogen on the farm. Our definition of outbreaks with sudden deaths encompassed all instances of outbreaks where chicken deaths were noticed without observation of any symptoms beforehand. Since farmers, or their family, check on their poultry at least once per day, it was assumed that these 'sudden deaths' corresponded to a time period of less than one day between onset of symptoms and death. For both the harvest and AI vaccination models, we assumed the effect of outbreaks on the dependent variable may be affected by the size of the considered flock ($n$). Consequently, we included this interaction term in the analysis.

The three dependent variables are likely affected by several farm-, flock-, and time-related factors, justifying the inclusion of several control covariates in the multivariable models, summarized in *Table 1*. For the harvest model, the main control variable is, logically, (1) the body weight of chickens, as broiler chickens are conventionally harvested after a fattening period upon reaching a given weight. Since the chicken weight was not collected during the survey, we used the difference between the current flock age $t$ and the anticipated age at maturity $t^*$ indicated by farmers in the questionnaire. Hereafter we use $\delta t = t - t^*$ for this difference. The shape of the function linking $\delta t$ and harvest may depart from linearity and is affected by the chicken breed, which determines the growth performance. Since information on chicken breed was not collected we used the age at maturity $t^*$ and the logarithm of flock size ($log(n)$) as proxy indicators of the growing performance of the breed and built a proxy body weight variable as a multivariate spline function of $\delta t$, $t^*$ and $n$ (*Burgos et al., 2008b*). 20% of flock-months had missing value for $t^*$. Since there was little within-farm variation in $t^*$ (2 months of difference at most between two flocks of the same farm), missing values were replaced by the median $t^*$ in the other flocks of the corresponding farm. (2) The calendar time $T$ was included as an additional smoothing spline term, since harvest may also be influenced by market prices which vary from one month to the next. Control variables included as standard

linear terms were (1) the number of chickens kept for laying eggs or breeding - famers with a large breeder-layer activity may want to keep some broilers chickens in the farm for replacing the breeding-laying stock, making them less likely to harvest broilers; (2) the number of broiler chickens simultaneously present in the same farm in other flocks; (3) the number of chicken flocks introduced in the same month; (4) the number of chicken flocks introduced in the previous month – farmers with a high number of broilers chickens or many recently introduced broiler flocks may want to sell their current flocks faster in order to limit feeding expenses and workload; (5) the vaccination status of the flock against AI; (6) the vaccination status of the flock against Newcastle Disease (ND) – farmers may keep their vaccinated flocks for a longer period as they are at lower risk of being affected by an infectious disease. We assumed the effect of outbreaks on the dependent variable may be affected by the size of the considered flock ($n$). Consequently, we included an interaction term between outbreaks and $log(n)$ in the analysis.

For the AI primo-vaccination model, control variables included as smoothing splines were (1) the flock age $t$ - vaccination may be preferentially done early in the flock life, (2) the flock size n, and (3) the calendar time $T$ - vaccination activities may be intensified at particular times of the year. Control variables included as standard linear terms were (1) the type of housing (free-range or confinement in pens or indoor) which affects the convenience of vaccination; (2) the proportion of the flock harvested in the same month - farmers might be less willing to vaccinate flocks being harvested; and the size of populations of (3) broiler chickens, (4) layer-breeder chickens, (5) broiler ducks, (6) layer-breeder ducks, (7) broiler Muscovy ducks and (8) layer-breeder Muscovy ducks kept in other flocks - farmers' perceived risk of AI and attitude towards vaccination may be influenced by the size of the poultry population at risk for AI and production type. We assumed the effect of outbreaks on the dependent variable may be affected by the size of the considered flock ($n$). Consequently, we included an interaction term between outbreaks and $log(n)$ in the analysis.

For the disinfection model, control variables included as smoothing splines were (1) the calendar time $T$ - disinfection activities may be intensified at particular times of the year. Control variables included as standard linear terms were the size of populations of (1) broiler chickens, (2) layer-breeder chickens, (3) broiler ducks, (4) layer-breeder ducks, (5) broiler Muscovy ducks and (6) layer-breeder Muscovy ducks - the farmers' attitude towards prevention may be influenced by the size of the poultry population at risk of disease.

## Multivariable modelling

We assumed that the events of interest, namely harvest, AI vaccination, and disinfection were drawn from a binomial distribution and used a logistic function to link their probability to a function of the independent covariates. Flocks were either fully vaccinated for AI or not at all, so the AI vaccination variable for flock-months took only the value 0 or one and was, therefore, treated as binary. Partial flock harvest (the harvest of only a fraction of the chickens in a given flock) and partial farm disinfection (the disinfection of facilities for only a fraction of the poultry flocks present in the farm) occurred in a minority of observations. Therefore, the number of chickens harvested per flock-month and the number of poultry flocks disinfected per farm-month were treated as binomial random variables with a number of trials equal to the flock size (for harvest) and the number of flocks per farm (for disinfection). To ensure that the model was not conditioned on the size of flocks and number of flocks per farm, prior weights equal to the inverse of the flock size and the number of flocks in the farm (i.e. the number of trials) were used in the binomial harvest model and disinfection model, respectively. The extent of over- or under-dispersion in the data was investigated by fitting a quasi-binomial model in parallel (*Papke and Wooldridge, 1996*). The resulting dispersion parameters were 0.76 (harvest model) and 0.77 (disinfection model), indicating moderate underdispersion, and that the estimates of our analyses are conservative.

Some of the included effects are non-linear in nature, and we needed to account for the intra-farm autocorrelation of the dependent variables. We therefore used a mixed-effects general additive model (MGAM) implemented in R with the 'mgcv' package (*Wood et al., 2016*). This enabled us to model the combined effect of $\delta t$, $t^*$, and flock size ($n$) on harvest time; the effect of $t$ and $n$ on AI vaccination; and the effect of calendar time ($T$) on all the dependent variables, as penalized thin plate regression splines (*Wood, 2017*). We specifically chose these variables because they are presumably the most important factors influencing the dependent variables and their effect could

possibly be highly non-linear. All other covariates were included as parametric regression terms. We also modelled the individual effects of farms on the dependent variables as random effects.

The complete models linking the logit $Y_{ij}$ of probability of realization of an event and the set of explanatory variables, for a flock-month $i$ (harvest, vaccination for AI) or a farm-month $i$ (disinfection) in a farm $j$, are described by the following set of equations:

Harvest model (flock-month level):

$$
\begin{aligned}
Y_{ij} &= \alpha + \sum_{m=0}^{2} \beta^{ONS-m} X_{ij}^{ONS-m} + \sum_{m=0}^{2} \beta^{OS-m} X_{ij}^{OS-m} \\
&+ f_{\delta t}\left(\delta t_{ij}, t_{ij}^{*}, log\left(n_{ij}\right)\right) + f_{T}\left(T_{ij}\right) + \sum_{k=1}^{6} \beta^{k} X_{ij}^{k} + \phi_{j} + \varepsilon_{ij}
\end{aligned}
\tag{1}
$$

AI vaccination model (flock-month level):

$$
\begin{aligned}
Y_{ij} &= \alpha + \sum_{m=0}^{2} \beta^{ONS-m} X_{ij}^{ONS-m} + \sum_{m=0}^{2} \beta^{OS-m} X_{ij}^{OS-m} + \sum_{m=0}^{2} \beta^{OD-m} X_{ij}^{OD-m} + \\
&\quad f_{t}\left(log\left(t_{ij}\right)\right) + f_{n}\left(log\left(n_{ij}\right)\right) + f_{T}\left(T_{ij}\right) + \sum_{k=1}^{8} \beta^{k} X_{ij}^{k} + \phi_{j} + \varepsilon_{ij}
\end{aligned}
\tag{2}
$$

Disinfection model (farm-month level):

$$
Y_{ij} = \alpha + \sum_{m=0}^{2} \beta^{O-m} X_{ij}^{O-m} + f_{T}\left(T_{ij}\right) + \sum_{k=1}^{6} \beta^{k} X_{ij}^{k} + \phi_{j} + \varepsilon_{ij}
\tag{3}
$$

The model parameters are $\alpha$ the model intercept; $\beta$ the parametric coefficients; $f$ a thin-plate spline function; $X^{k}$ the general notation for variables with linear effects; $X^{O-m}$, $X^{OS-m}$, $X^{ONS-m}$ and $X^{OD-m}$, categorical variables denoting presence or absence of an outbreak in the same farm $m$ months prior in any species (O), in chickens with sudden deaths (OS), in chickens with no sudden deaths (ONS), and in different species (OD) respectively; $n$ the flock size; $t$ the current age of the flock; $t^{*}$ the age at maturity of the flock anticipated by the farmer; $\delta t$ the difference between current age and age at maturity; $T$ the calendar time; $\varphi$ the farm random effect; $\varepsilon$ the residual error term. Some variables with a highly skewed distribution (*Table 1*) were transformed. Current age ($t$) and flock size ($n$) being strictly positive, they were log-transformed. Farm populations of broiler and layer-breeders of different species being null or positive, they were square-root transformed. Covariates included in the multivariate spline function for body weight ($\delta t$, $t^{*}$, $log(n)$) were centered and standardized. Interaction terms between outbreak categorical variables and flock size $log(n_{ij})$ were added in the Harvest and AI vaccination models.

Excessive multi-collinearity between covariates was assessed by estimating their variance inflated factor using the 'usdm' R package (*Naimi et al., 2014*). We fitted the complete models using the whole set of covariates using restricted maximum likelihood estimation. We then used a backward-forward stepwise selection, based on AIC comparison, to eliminate the variables with non-significant effects (*Hosmer and Lemeshow, 2000*).

Arguably, one farmer is likely to maintain the same farm management from one month to the next despite changes in influential covariates. Therefore, for each model, we tested the presence of farm-level temporal autocorrelation by fitting two linear regression models on the deviance residuals, with a fixed constant effect and with and without intra-farm AR-1 time autocorrelation structure and comparing the two model fits with a log-likelihood ratio test. For the 'disinfection' model, the fit was significantly improved by including the autocorrelation term while for the two other models it was not. Therefore, we implemented the same model fitting protocol for the 'disinfection' model with an additional intra-farm AR-1 time autocorrelation term on the dependent variable. We used the 'gamm' routine of the 'mgcv' package for this purpose (*Wood, 2017*). Since 'gamm' models for binomial data are fitted with the penalized quasi-likelihood approach, the AIC metric is not suitable to compare such models. Instead, we implemented a stepwise removal of covariates whose t-test returned the highest probability of type one error (p-value) until all remaining covariates had a p-value lower than 20%.

All analyses and graphical representations were performed with R version 3.6.1 (*R Development Core Team, 2014*).

## Ethical statement

The collaboration between the investigators (authors) and the Ca Mau sub-Department of Livestock Production and Animal Health (CM-LPAH) was approved by the Hospital for Tropical Diseases in Ho Chi Minh City, Vietnam. The CM-LPAH, which at the province-level is the equivalent of an ethical committee for studies on livestock farming, specifically approved this study.

## Acknowledgements

Funding: The study was supported by the Defense Threats Reduction Agency (US), by Wellcome Trust grant 098511/Z/12/Z, and by Pennsylvania State University.

## Additional information

### Funding

| Funder | Grant reference number | Author |
| --- | --- | --- |
| Wellcome | 098511/Z/12/Z | Maciej F Boni |
| Defense Threat Reduction Agency | | Maciej F Boni |
| Pennsylvania State University | | Maciej F Boni<br>Alexis Delabouglise |

The funders had no role in study design, data collection and interpretation, or the decision to submit the work for publication.

### Author contributions

Alexis Delabouglise, Data curation, Formal analysis, Visualization, Methodology, Writing - original draft; Nguyen Thi Le Thanh, Huynh Thi Ai Xuyen, Investigation, Project administration; Benjamin Nguyen-Van-Yen, Data curation, Writing - review and editing; Phung Ngoc Tuyet, Project administration; Ha Minh Lam, Project administration, Writing - review and editing; Maciej F Boni, Supervision, Validation, Project administration, Writing - review and editing

### Author ORCIDs

Alexis Delabouglise  https://orcid.org/0000-0001-5837-7052
Maciej F Boni  http://orcid.org/0000-0002-0830-9630

### Ethics

Human subjects: The research collaboration was approved by the Hospital for Tropical Diseases in Ho Chi Minh City, and the study design was approved by the Ca Mau sub-Department of Livestock Production and Animal Health. The Ca Mau sub-Department of Livestock Production and Animal Health (CM-LPAH) specifically approved this study and is equivalent to an Animal Care and Use Committee that approves studies like this in Vietnam. CM-LPAH approved the publication of these results. No consenting process was required as the study involved no human biological samples, no patients, and no personal or identifiable information. The IRB that made this determination was the Hospital for Tropical Diseases Scientific and Ethical Committee (Ho Chi Minh City).

### Decision letter and Author response

Decision letter https://doi.org/10.7554/eLife.59212.sa1
Author response https://doi.org/10.7554/eLife.59212.sa2

## Additional files

### Supplementary files

- Supplementary file 1. Fitted parameters of the original broiler chicken harvest model.

- Supplementary file 2. Fitted parameters of the broiler chicken harvest model with aggregated effects of outbreaks with and without sudden deaths.
- Transparent reporting form

## Data availability

The study dataset is available online at the Open Science Framework, https://osf.io/ws3vu/.

The following dataset was generated:

| Author(s) | Year | Dataset title | Dataset URL | Database and Identifier |
|-----------|------|---------------|-------------|-------------------------|
| Delabouglise A, Boni M | 2020 | Poultry Population and Farm Management Dynamics - Ca Mau - Viet Nam | https://doi.org/10.17605/ OSF.IO/WS3VU | Open Science Framework, 10.17605/ OSF.IO/WS3VU |

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
