## [Decision Letter]

**Acceptance summary:**

Your manuscript identifies that depopulation is a more common response to disease outbreaks among farms with small flocks in southern Vietnam. This may prove to be a vital piece of information for policies that aim to limit the spread of avian influenza into human populations.

**Decision letter after peer review:**

Thank you for submitting your article "Poultry farmer response to disease outbreaks in smallholder farming systems" for consideration by *eLife*. Your article has been reviewed by three peer reviewers, and the evaluation has been overseen by a Reviewing Editor and Miles Davenport as the Senior Editor. The following individuals involved in review of your submission have agreed to reveal their identity: Benny Borremans (Reviewer #1); Cassidy Rist (Reviewer #2); Andrés Garchitorena (Reviewer #3).

The reviewers have discussed the reviews with one another and the Reviewing Editor has drafted this decision to help you prepare a revised submission.

Summary:

Your manuscript surveyed 53 poultry farms in Southern Vietnam and identified that small scale farmers with lower sized flocks were more likely to rapidly harvest and sell disease birds to mitigate loss of profit. This finding is of great potential importance for developing prevention efforts for introduction of avian influenza into human populations.

Essential revisions:

The reviewers were all highly complementary of this paper. They all felt the manuscript was methodologically sound, clearly written, highly original and of substantial public health and policy relevance. Particular noted strengths were appropriate use and description of mixed-effects general additive models, appropriate study design and the inclusion of all raw data for public use. The following suggestions will help clarify your scientific messaging:

First, I suggest a title with more specific language such as "Poultry farmer response to avian influenza outbreaks in smallholder farming systems Southern Vietnam"

The Discussion could use a more detailed limitations section. For instance, please elaborate on 1) the use of a proxy for weight instead of weight itself, 2) potential misclassification of outbreaks, 3) the fact some behaviours may depend on events happening in longer time frames, for example the previous year, but this is not accounted for in the models, and 4) that the harvest of chickens could be greatly influenced by economic needs of the household (a family event, an economic shock, disease, etc.), especially for smallholder farmers in the developing world who may use chickens as a form of cash savings.

The authors should also expand the practical implications of the study in terms of policies or interventions to put in place. The current version of the Discussion lacks these insights. Given the results, what can the government or NGOs or international organizations implement in order to reduce the risk of future outbreaks? This section should be expanded and be more specific.

---

## [Author Response]

Essential revisions:The reviewers were all highly complementary of this paper. They all felt the manuscript was methodologically sound, clearly written, highly original and of substantial public health and policy relevance. Particular noted strengths were appropriate use and description of mixed-effects general additive models, appropriate study design and the inclusion of all raw data for public use. The following suggestions will help clarify your scientific messaging:First, I suggest a title with more specific language such as "Poultry farmer response to avian influenza outbreaks in smallholder farming systems Southern Vietnam"

Thank you for this suggestion. We agree the title needs to be more specific. However, since outbreaks in our study are defined by the observation of poultry deaths with clinical symptoms (which, in some instance, correspond to suspicion – but not confirmation – of avian influenza), a title suggesting that we captured avian influenza outbreaks would be misleading. We propose the following title instead: “Poultry farmer response to disease outbreaks in smallholder farming systems in southern Vietnam”.

Please note that the ‘s’ in southern Vietnam needs to be lowercase, as two of the authors are government employees in Vietnam; ‘southern Vietnam’ is a region of Vietnam but ‘Southern Vietnam’ implies that it is an administrative entity, which it is not.

The Discussion could use a more detailed limitations section. For instance, please elaborate on 1) the use of a proxy for weight instead of weight itself, 2) potential misclassification of outbreaks, 3) the fact some behaviours may depend on events happening in longer time frames, for example the previous year, but this is not accounted for in the models, and 4) that the harvest of chickens could be greatly influenced by economic needs of the household (a family event, an economic shock, disease, etc.), especially for smallholder farmers in the developing world who may use chickens as a form of cash savings.

These are relevant comments. We completed the Discussion accordingly (fourth, sixth and seventh paragraphs).

The authors should also expand the practical implications of the study in terms of policies or interventions to put in place. The current version of the Discussion lacks these insights. Given the results, what can the government or NGOs or international organizations implement in order to reduce the risk of future outbreaks? This section should be expanded and be more specific.

This is a very good point, not very well developed in the initial manuscript, indeed. We added a paragraph to the Discussion (eighth paragraph).